# The Influence of the Structure Parameters on the Mechanical Properties of Cylindrically Mapped Gyroid TPMS Fabricated by Selective Laser Melting with 316L Stainless Steel Powder

**DOI:** 10.3390/ma15124352

**Published:** 2022-06-20

**Authors:** Tomasz Szatkiewicz, Dorota Laskowska, Błażej Bałasz, Katarzyna Mitura

**Affiliations:** Faculty of Mechanical Engineering, Koszalin University of Technology, Śniadeckich 2, 75-620 Koszalin, Poland; blazej.balasz@tu.koszalin.pl (B.B.); katarzyna.mitura@tu.koszalin.pl (K.M.)

**Keywords:** additive manufacturing, triply periodic minimal surfaces, energy absorption, quasi-static compression

## Abstract

The development of additive manufacturing techniques has made it possible to produce porous structures with complex geometry with unique properties as potential candidates for energy absorption, heat dissipation, biomedical, and vibration control application. Recently, there has been increased interest in additively manufacturing porous structures based on triply periodic minimal surfaces (TPMS) topology. In this paper, the mechanical properties and energy absorption abilities of cylindrical mapped TPMS structures with shell gyroid unit cells fabricated by selective laser melting (SLM) with 316L stainless steel under compression loading were investigated. Based on the experimental study, it was found that tested structures exhibited two different deformation modes. There is also a relationship between the number and shapes of unit cells in the structure and the elastic modulus, yield strength, plateau stress, and energy absorption. These results can be used to design and manufacture more efficient lightweight parts lattices for energy absorbing applications, e.g., in the field of biomedical and bumpers applications. The deformation mode for each tested sample was also presented on the records obtained from the ARAMIS system.

## 1. Introduction

Porous structures are a universal term used to describe the size, distribution, and morphology of pores of the materials. They can be classified by porosity types (closed and open pores) and by the arrangement of elementary cells (stochastic and non-stochastic) [1]. Traditional manufacturing methods of porous structures are controlled powder sintering, polymeric sponge replication, molding and sintering of short metal fibers, and solid-state foaming by an expansion of argon-filled pores. These techniques have many limitations, such as contamination, impurity phases, or limited and predetermined part geometries. However, the biggest challenge is limited control over pore size, shape, volume fraction, and their spatial distribution [1,2,3]. Additive manufacturing (AM) may overcome these limitations. Moreover, with a wide range of available technologies, it is possible to produce porous structures from various materials, including thermoplastics, resin, ceramics, metals, and alloys.

Recently, there has been increased interest in additively manufacturing porous structures based on triply periodic minimal surfaces (TPMS) topology. The TPMS are naturally inspired structures with periodically, infinite continuous non-self-intersecting surfaces with zero mean curvature in three independent directions. Porous architectures with the TPMS topology are constructed by repeating elements, so-called unit cells [4,5,6]. The main advantages of TPMS-type structures are elimination of the effect stress concentration at nodal points due to the continued curvature of the cells walls [7], relatively large surface area, continuous internal channels [4], and self-supported architecture [8]. Moreover, by changing parameters such as relative density, porosity, pore size, or wall thickness, it is possible to design structures with expected mechanical properties. TPMS structures also have a good fatigue behavior [9]. Therefore, the TPMS structures are potential candidates for energy absorption [10,11,12,13], heat dissipation [14], biomedical [3,15,16,17,18,19], and vibration control applications [20].

Due to of complex architecture of TPMS structures, its fabrication using powder bed fusion techniques (such as selective laser melting or electron beam melting) is a challenge. The process parameters and design should be selected taking into account manufacturing constraints such as printing resolution, removal of necessary supports, and internal powder entrapment [21,22,23].

The properties of TPMS lattice structures are of interest to many researchers. According to Mahmoud et al. [24], the factors affecting the mechanical properties of lattice structures can be categorized into manufacturing and design-related factors. Most authors focus on the impact of the design-related factors, such as relative density (volume fraction), porosity, pore size, and cell type on mechanical behavior and energy absorption capacity under compressive loading conditions, by comparing different types of structures with uniform or graded morphology. The theoretical basis for describing the mechanical response of porous (lattice) structures under compressive load was developed by Gibson et al. [25].

Zaharin et al. [1] investigated the influence of unit cell type (cube and gyroid) and pores size (range of 300 to 600 µm) on porosity, and mechanical behavior additively manufactured Ti6Al4V porous structure. Research showed that the elastic modulus and yield strength of the analyzed samples decrease with an increase its porosity. The obtained values for gyroids were similar to human bones, which confirms the usefulness of these TPMS structures in applications in bone tissue engineering and orthopedics. Yan et al. [8] indicated that Young’s modulus and yield strength of gyroid cellular lattice structures increases with the increase in its volume fraction (relative density).

Maskery et al. [26] accentuate that the deformation mode and the related stress–strain curve (especially the long and flat plateau) determine the suitability of the structure in energy absorption applications. They investigated the failure models in double (matrix phases) gyroid structures made of Al-Si10-Mg alloy and fabricated using selective laser melting technology. The samples differed in unit cell size, respectively, 3, 4.5, 6, and 9 mm, while maintaining the same relative density (volume fraction). The tested samples showed three types of failure under compressive load: “layer-by-layer” mode, brittle fracturing of the cell walls mode with the propagation of a crack (or cracks) through the lattice structure, and diagonal shear mode. The failure mode depends on the cell size. Moreover, they showed that a way to change deformation behavior in the case of Al-Si10-Mg samples is post-manufacture heat treatment.

Special attention should also be noted to the works that evaluated the mechanical properties, failure resistance, and permeability of different types of TPMS structures. Yanez et al. [27] investigated the diamond and gyroid structures that were fabricated using electron beam melting (EBM) technology from Ti-6Al-4V alloy and observed that specific strength (compressive strength against density) for gyroid structures is correlated with strut (wall) angle. Decreasing the strut angle increased the modulus of elasticity and compressive strength. Moreover, they concluded that the gyroid structures showed better strength to weight ratios in comparison with other TPMS structures. Bobber et al. [28] evaluated the three types of TPMS structures, primitive, diamond, and gyroid, fabricated using laser selective melting technology from Ti-6Al-4V alloy. For every unit cell, the porosity and surface area decreases with increasing wall thickness. The tested samples also showed exceptionally high resistance to fatigue and a unique combination of relatively low modulus of elasticity and high yield point. The mechanical properties of primitive, diamond, and gyroid structures were also investigated by Maskery et al. [29]. TPMS lattice structures were fabricated using selective laser sintering from polyamide. Research shows that the sample’s deformation process, failure mode, and mechanical properties depend on cell geometry. The primitive lattice structure shows the highest elastic modulus, but the deformation process was characterized by structural bucking and low failure strain. The authors indicate that this may be related to stretching-dominated deformation. In turn, the gyroid and diamond structures showed quite similar mechanical responses typical for a structure in which the deformation process is dominated by bending. These conclusions were also reflected in the research conducted by Yang et al. [5]. Moreover, based on numerical and experimental results, Yang et al. [5] concluded that geometrical factors affecting on stiffness and strength of analyzed cubic gyroid samples are the number of cells, surface thickness, bulk size, and isovalue. Zhang et al. [12] investigated the energy absorption capacity of gyroid, diamond, and primitive TPMS sheet structures. The experimental results show that the mechanical response and deformation mechanism of the tested structure depends on unit cell geometry. The primitive shape samples presented a collapse mechanism by diagonal shear after yielding, while the diamond and gyroid shapes samples presented stable collapse mechanisms due to relatively uniform stress distributions across unit cells. Based on the results authors concluded that diamond structures have the largest stiffness and energy absorption ability.

In the literature, much attention is also paid to the mechanical property of TPMS structures with uniform or graded morphology. Li et al. [30] compared the mechanical properties and energy absorption of sheet-based (matrix phases) and strunt-based (network phases) gyroid structures with uniform and graded density fabricated using stereolithography technology. Based on numerical homogenization results, it was observed that sheet-based gyroid structures have a higher elastic modulus in comparison with strunt-based gyroids (at the same volume fraction). These observations were also confirmed by experimental results. Moreover, the samples presented the different collapse deformation models: for the uniform samples—the global collapse, and the graded samples—the layer-by-layer collapse. The result also shows that the energy absorption capacity of the graded structure is better than that of the uniform structure. Additionally, the sheet-based gyroid structure has better energy absorption than the strunt-based gyroid structure. Yang et al. [31] investigated the mechanical properties of uniform and continuous graded gyroid cellular structures fabricated by selective laser melting. Experimental compression results indicated that the deformation process depends on the density gradient (perpendicular or parallel) to the loading direction. Moreover, the graded gyroid samples exhibit improved mechanical properties compared with uniform gyroid samples. Mahomud et al. [9] tested the mechanical properties of gyroid samples with three different designs: uniform porosity with thin and thick walls and graded porosity. Research shows that the compressive strength of samples with graded porosity was higher than uniformly samples. Research on the mechanical properties of graded TPMS structures shows that they can be more beneficial for loading-bearing applications.

In energy absorption applications, cylindrical structures are most commonly used. In order to reduce their weight, which is important in transport, aerospace, and cosmonautics, the traditional cylindrical structures with solid materials began to be replaced by lattice structures. Cylindrical lattice structures (CLS) consist of ribs in the circumferential and helical directions, and the crossing of the ribs creates periodic patterns [32]. These solutions show unique loads paths and are characterized by high durability, height stiffness-to-weight characteristics, and good energy absorption properties, which were demonstrated, inter alia, in the works of Shitanaka et al. [32], Gu et al., [33], Smeets et al. [34], Cao et al. [35], and Meng et al. [36].

Thanks to additive manufacturing, the construction of straight ribs in CLS can be replaced by complicated spatial structures based on TPMS topology, as was indicated by Wang et al. [37].

In this paper, the mechanical properties and energy absorption abilities of cylindrical mapped TPMS structures with shell gyroid unit cells fabricated by selective laser melting (SLM) with 316L stainless steel under compression loading were investigated. The aim of the research was to find the relationship between design parameters of the sample structure and elastic modulus, yield strength, plateau stress, and total energy absorption per unit volume. An attempt was also made to analyze the deformation modes and correlate them with the course of the stress–strain curve and the efficiency–strain curve.

## 2. Materials and Methods

### 2.1. Theoretical Background

For a better understanding, it is necessary to introduce the appropriate nomenclature regarding cellular (porous, lattice) structure design. According to the Gibson–Ashby model [24], one of the most important factors that affect the mechanical properties of cellular structures is the relative density, defined as the ratio of the density of the lattice structure (ρ_L_) to the density of the base solid material (ρ_S_):ρ^*^ = (ρ_L_/ρ_S_) × 100%.(1)

Similarly, the relative elastic modulus is defined as the ratio of the elastic modulus of the lattice structure (E_L_) to the elastic modulus of the solid material (E_S_):E^*^ = E_L_/E_S_.(2)

Both values are linked by the equation:E^*^ = C_1_ × ρ^*n^,(3)
where C_1_ is the coefficient as the range of values from 0.1 to 4.0 and n is constant with values of approximately 2.

The Gibson–Ashby model also describes the relationship between relative density, plateau stress σ_L_, and densification strain ε_D_:σ_L_ = C_2_ × ρ^*m^ × σ_S_,(4)
ε_D_ = 1−αρ^*^,(5)
where C_2_ is the coefficient as the range of values from 0.25 to 0.35, m is constant with values of approximately 3/2, and α depends on the matrix material deformation behavior. The real value of the constant parameters C_1_, C_2_, n, m, and α are calculated based on the results of the compression test [37,38].

As demonstrated by Gibson et al. [24], the cellular structures showed interesting mechanical behavior under compressive testing. The uniaxial stress–strain curve (Figure 1) consists of a linear elastic, plateau stress, and densification section. The linear elastic section is related to bending for inclined cell walls and stretching for vertical cell walls. The elastic modulus of the lattice structure corresponds to the tangent of the inclination angle of this section of the curve to the strain axis. The plateau section is related to the creation of plastic hinges at the sections or joints of the cell walls. For plastic materials, it is a nearly flat section of the curve characteristic of approximately constant stress. The brittle materials exhibit a fluctuation around a stress value in this section. After crossing strain ε_D_ (also called the densification point), the structure enters the densification section, where the individual cell walls start to collapse. In this way, the stress is transferred throughout all cellular structures (not only cell walls), which results in dramatically increased strength and, finally, damage to the object.

Based on the uniaxial stress–strain curve, the energy absorption efficiency can be defined as energy absorption divided by the product of the maximum compressive stress within the strain range and the magnitude of the strain range [30,31,39,40]:η(ε) = (1/σ(ε)) × W_V_,(6)
where W_V_ is energy absorption capacity defined as the area under the stress–strain curve (energy absorption per unit volume):W_V_ = ∫σ(ε)dε.(7)

The densification point is considered to be the practical limit for energy absorption applications using cellular structures and is given by:dη(ε)/dε|ε = ε_D_ = 0.(8)

### 2.2. Gyroid Lattice Structure Design

The gyroid unit cell in one of the most common types of TPMS, which was discovered in the 1970s by A. Schoen. Gyroid can be generated by finding the isosurface (U = 0) of following equation [37,38]:
U = [sin(k_x_,x)cos(k_y_,y) + sin(k_y_,y)cos(k_z_,z) + sin(k_z_,z)cos(k_x_,x)]^e^ − t^e^,(9)
where $k_{i}$ are TPMS function periodicities, defined by:k_i_ = 2π * u_i_/L_i_  i = x,y,z,(10)
where u_i_ are the number of cell repetitions in x, y, and z, and L_i_ are the absolute size of the structure in those dimensions.

Depending on the value of the exponent e, there are two types of gyroid unit cell (Figure 2):
Network structure (for e = 1), which contains two continuous regions of space (solid and void) separated by isosurface (U = 0);Matrix structure (for e = 2), which contains three regions (a wall of solid material bounded by two unconnected void regions). This type of structure has also been referred to as “shell”, “sheet”, “strut”, or “cellular” in the literature.

The t parameter in Equation (9) determines the volume fractions pertaining to the regions separated by surface [6,8,30,38].

In this study, nTopology (nTopology, New York, NY, USA) software [41] was utilized to design the cylindrical samples with 30 mm of outer diameter, 18 mm of inner diameter, and 24 mm of height. The essence of the sample design process was to obtain cylindrical mapped TPMS porous structures with shell gyroid unit cells. The cubic unit cell was a transformation to sectorial shape through to converting the Cartesian coordinate system to a polar coordinate system. In order to generate a lattice structure, the sectorial unit cell was circumferentially arrayed with a center point of the coordinate origin, as shown in Figure 3.

In this study, the lattice structures had 12, 9, and 6 cells in a circumferential direction (n_circum_), 1, 1.5, and 2 cells in the radial direction (n_radial_), and 3 cells in an axial direction (n_axial_). The wall thickness T was chosen such that the relative density of each sample was 20%. Therefore, a series of 9 samples with different architecture was designed. For recognition of the samples, a name coding system was introduced, in which consecutive terms mean: type of TPMS topology, number of unit cells in a circumferential direction, number of unit cells in a radial direction, relative density, and wall thickness.

### 2.3. Powder Characterization and Sample Fabrication

The samples were made from 316L austenitic stainless steel powder with an average particle size of 45 ± 15 µm, which was produced in a gas atomized process by Oerlikon Metco Inc., Troy, MI, USA (MetcoAdd^TM^ 316L-A). The chemical components of the powder are shown in Table 1. The scanning electron microscope (SEM) image (Figure 4) shows a variety of grain sizes and morphology.

The samples were fabricated using an ORLAS CREATOR^®^ (O. R. Lasertechnologie GmbH, Dieburg, Germany) selective laser melting (SLM) machine. The SLM technique uses a laser as the source of thermal energy to fuse a selected area of the powder. The typical SLM machine consists of an enclosed chamber with a laser, powder bed (building platform), powder bed supply, re-coater arm, and inter gas system. The role of the re-coater arm is to feed a new excellent packed powder layer to the building platform without disturbing the previous layers. Next, the powder was laser scanned and fused according to the CAD project. After the layer was printed, the build plate was lowered. These steps were repeated until a complete print was obtained. In the SLM process, an important is the printing environment. In order to prevent oxidation during the process, the work chamber must be infilled with protective gas [21,22,23,43].

The processing parameters such as laser power, laser speed, and layer thickness are shown in Table 2. The supports connecting the printout with the build plate were removed mechanically. In the case of samples with n_radial_ = 1.5, other supports were also used (Figure 5).

After fabrication, each sample was weighed on a precision balance scale with an elementary plot d = 0.01 g. Designed mass (m_d_) was calculated by the equation:m_d_ = ρ_S_ * V_L_,(11)
where V_L_ is the volume of cellular structure calculated based on the CAD project. In this study, the density of the base, solid material ρ_S_, was set as 7.578 g/cm^3^ based on measurements from the Archimedean balance of the cubes (dimensions 20 × 20 × 20 mm) fabricated from MetcoAddTM 316L-A powder (Oerlikon Metco Inc., Troy, MI, USA) using an ORLAS CREATOR^®^ SLM machine (O. R. Lasertechnologie GmbH, Dieburg, Germany). Based on this, it was established that mass of samples with relative densification of 20% amount to 16.46 g. If the difference between the design mass and the real mass (m_r_) of the samples was less than 1%, then the samples qualified for the quasi-static compression test.

Therefore, a series of 27 samples with different architecture was designed and fabricated (Figure 6). The details of the samples studied in this paper are provided in Table 3.

### 2.4. Quasi-Static Cmompession Test

Quasi-static compression tests were performed on a Zwick Z400E (ZwickRoell GmbH and Co., Ulm, Germany) machine. The tests were conducted according to the ISO 133:14 standard [44]. The samples were compressed at a strain rate of 2 mm/min according to the production direction. For each design, three tests were carried out to enhance the accuracy of the results. During the tests, the image was archived using the ARAMIS (GOM GmbH, Brunswick, Germany) measurement system. The obtained images were analyzed using the software GOM Correlate 2020 (GOM GmbH, Brunswick, Germany).

Based on stress–strain curvature for each sample, the elastic modulus, yield strength, plateau stress, and energy absorption capacity were obtained. The elastic modulus was defined as the slope of the linear elastic section of the curve, and the yield strength was determined at 0.2% offset strain. According to ISO 13314: 2011, the plateau stress was defined as the arithmetical mean of stress at strain intervals between 20% and 30% compressive strain.

## 3. Results and Discussion

### 3.1. Compression Test Results

A representative stress–strain curves from compressive testing of the samples are presented in Figure 7, Figure 8 and Figure 9. The curves obtained for samples with the same specification were quite similar, which indicates high repeatability of the manufacturing process. All curves exhibit three sections typical for cellular structures described by Gibson–Ashby model: linear elastic section, followed by a long stress plateau, and ended by densification section. Before the linear elastic section, a nonlinear stage occurred because, during the compression, a full-contact condition between the samples and the crosshead was established [17].

The analysis of the stress–strain curves (Figure 7, Figure 8 and Figure 9) showed a relationship between the course of the plateau region and the number of unit cells in a radial direction (n_radial_). An increase in n_radial_ resulted in the appearance of oscillations in the plateau region. The largest oscillations were observed for the sample Gyroid_12_2_20_0.58. By comparing the plateau region course of all samples with n_radial_ = 2, it can be observed that reducing the number of unit cells in the circumference direction (n_circum_), and thus increasing the wall thickness (T), decreased the amount and intensity of the oscillations. For the remaining samples, a relatively flat course of the plateau area was observed.

The elastic modulus E_L_, yield strength σ_y_, and plateau stress σ_L_ (mean and standard deviation) of the cylindrical mapped gyroid structures obtained from the stress–strain curves are shown in Table 4. The data are also presented in Figure 10. Based on the data, it was found that reducing n_circum_ and n_radial_ reduces the elastic modulus, yield strength, and plateau stress.

### 3.2. Energy Absorption

Figure 11 presents the cumulative energy absorption per unit volume plotted against the effective lattice strain. After crossing strain ε_D_ (also called the densification point), the structure enters the densification section in which the individual cell walls start to collapse. Thus, the stress is transferred throughout all cellular structures (not only cell walls), which results in dramatically increased strength and, finally, damage to the object. Therefore, the densification point is the limit of the suitability of a given structure for energy absorption applications. As Figure 11 shows, all samples showed linear energy absorption with strain until the point of densification was reached.

Table 5 presents the values of densification points ε_D_ and total energy absorbed per unit volume up to this strain W_V_ for all tested samples. The data are also presented in Figure 12.

The highest value of densification point (56%) was observed in the case of the Gyroid_9_2_20_0.64 sample, and the lowest value (47%) in the case of Gyroid_6_1.5_20_0.80. The highest value of the total energy absorbed per unit volume up to the densification point (19.26 MJ/m^3^) was observed in the case of Gyroid_9_1_20_0.77, and the lowest value (9.72 MJ/m^3^) in the case of Gyroid_6_2_20_0.73 sample.

Based on the data, it was found that for samples with n_circum_ = 9 and n_circum_ = 6, increasing the number of a unit cell in the radial direction causes reduces the total energy absorbed per unit volume up to the densification point. In the case of samples with n_circum_ = 12, the total energy absorbed up to the densification point was similar for all samples.

Figure 13 presents the efficiency–strain curve. In the course of the efficiency–strain curve, two stages can be distinguished: the rising stage, which is bound by a gradual increase in the energy absorption efficiency, and the declining stage, which is bound by a gradual decrease in the energy absorption efficiency. The border between these stages is the densification point. The lack of oscillation in the rising stage proves that the samples presented the stability of absorption energy during compression [39,40].

### 3.3. Compressive Deformation Behavior

The data for the course of the efficiency–strain and stress–strain curves were compared with images obtained with the ARAMIS measurement system to analyze the deformation mode of tested structures. Based on this, it was found that the tested samples exhibited two different deformation modes.

Figure 14A shows the global uniform deformation mode, which is characterized by buckling and folding of the cell walls in the middle part of the structure [37]. This type of failure gave the stress–strain cures with a relatively flat plateau (Figure 14B) and no oscillation in the rising stage of the efficiency–strain curve, with an explicit maximum at the densification point (Figure 13). Global uniform mode was presented by Gyroid_12_1_20_0.70 and Gyroid_9_1_20_0.77 samples.

The second deformation mode is characterized by a successive collapse of cells in planes perpendicular to the manufacturing and loading direction [26]. This is the so-called “layer-by-layer” deformation mode. Two variants of this deformation mode were observed:
A variant with complete densification of collapsing layers. In this case, the oscillations in the plateau region of stress–strain curve were observed (Figure 15B). Maximums and minimums in the raising region of the efficiency–strain curve indicate the moment of densification of the layer. This variant of “layer-by-layer” deformation mode was observed for Gyroid_6_1_20_0.87 (Figure 15A);A variant with incomplete densification of simultaneously or successively collapsing layers. In this case, no oscillation in the plateau region of the stress–strain curve was observed (Figure 16B). In the course of raising the region of the efficiency–strain curve, there were no maximums and minimums. There is also no clear maximum at the densification point (Figure 13). This variant of “layer-by-layer” deformation mode was presented for an example of Gyroid_6_2_20_0.73 (Figure 16A).

A special case of deformation mode was presented for the Gyroid_12_2_20_0.58 (Figure 17A). It was distinguished from other “layer-by-layer” deformation modes by distinct oscillations in the plateau region of stress–strain curve, in the range of strain ε = 0–25%. Over 25% of strain, the sample presented the “layer-by-layer” collapse mode with incomplete densification of layers. The specific behavior of the sample may be related to the fact that its wall thickness had the smallest value (0.58 mm) of all tested samples. It should also be noted that this value was close to the technological limit.

Table 6 shows an attempt to classify the tested samples to the observed deformation modes. The deformation mode for each tested sample was also presented on the records obtained from the ARAMIS system (Appendix A). It is difficult to indicate a clear trend of the variability of the deformation mode with the change in design parameters (n_radial_ and n_circus_) in the conducted range of the experiment. According to this, the authors of the study think that for a better understanding of the deformation modes and in order to indicate the usefulness of the studied structures for the proposed applications (energy absorption), it would be beneficial to extend the experiment by, among others, increasing the range of variability of design parameters (nr_adial_ and _ncircum_) or influence others design parameters (n_axial_, internal diameter, wall thickness).

## 4. Conclusions

In this paper, the mechanical properties and energy absorption abilities of cylindrical mapped TPMS structures with shell gyroid unit cells fabricated by selective laser melting (SLM) from 316L stainless steel under compression loading were investigated. The conducted experiment is an extension of the experimental work carried out by Wang et al. [37].

Based on the stress–strain curve, the following relationships between the design parameters of the samples and the course of the plateau area were observed:An increase in the number of unit cells in the radial direction (n_radial_) causes appearances oscillations in the plateau region;For samples with n_radial_ = 2, reducing the number of unit cells in circumference direction (n_circum_) and thus increasing the wall thickness (T) causes a decrease in the amount and intensity of the oscillations;for samples with n_radial_ = 1 and n_radial_ = 1.5, a relatively flat course of the plateau area was observed.

It was observed that the value of elastic modulus, yield strength, and plateau stress of tested samples also depends on the design parameters:An increase in the number of unit cells in the circumferential direction (n_circum_) causes an increase in the value of E_L_, σ_y_, σ_L_;A decrease in the number of unit cells in the radial direction (n_radial_) causes an increase in the value of E_L_, σ_y_, σ_L_.

In the field of energy absorption applications, it is important to maximize the total energy absorption and densification point. Based on the data, it was noticed that:For samples with n_radial_ = 1.5 i n_radial_ = 2, an increase in the number of unit cells in the circumferential direction (n_circum_) causes an increase in the value of total energy absorption per unit volume up to the densification point;for samples with n_radial_ = 1, there is no such clear relationship; however, the sample with n_circum_ = 6 absorbed the least amount of energy;for samples with n_circum_ = 9 and n_circum_ = 6, a decrease in the number of unit cells in the radial direction (n_radial_) causes an increase in the value of total energy absorption per unit volume up to the densification point;for samples with n_circum_ = 12, there is no such clear relationship.

For the tested samples, the highest value of densification point (56%) was observed for the Gyroid_9_2_20_0.64 sample and the lowest (47%) for the Gyroid_6_1.5_20_0.80 sample.

The highest value of the total energy absorbed per unit volume up to the densification point (19.26 MJ/m^3^) was observed for the Gyroid_9_1_20_0.77 sample. This sample also had one of the highest densification point values (54%). The deformation mode of this sample was classified as global uniform. The lowest value of the total energy absorbed per unit volume up to the densification point (9.72 MJ/m^3^) was observed for the Gyroid_6_2_20_0.73. This sample ranks 5th in terms of the densification point value (52%). The deformation mode of this sample was classified as “layer-by-layer” with incomplete densification of collapsing layers.

In conclusion, a relationship between the design parameters (n_radial_ i n_circum_) and mechanical properties (E_L_, σ_y_, σ_L_) and energy absorption ability in the analyzed range of variability of design parameters was observed. It is difficult to indicate a clear trend of the variability of the deformation mode with the change in design parameters (n_radial_ and n_circus_). According to this, the authors of the study concluded that for a better understanding of the deformation modes phenomenon and to indicate the usefulness of the studied structures for the proposed applications (energy absorption), it would be beneficial to extend the experiment by, among others:Increasing the range of variability of design parameters (n_radial_ and n_circum_);Analysis of the impact of changing the internal diameter;Analysis of the effect of changing the number of unit cells (layers) in the axial direction (n_axial_);Analysis of the effect of the gradient of the wall thickness change in the axial direction.

## Figures and Tables

**Figure 1 materials-15-04352-f001:**
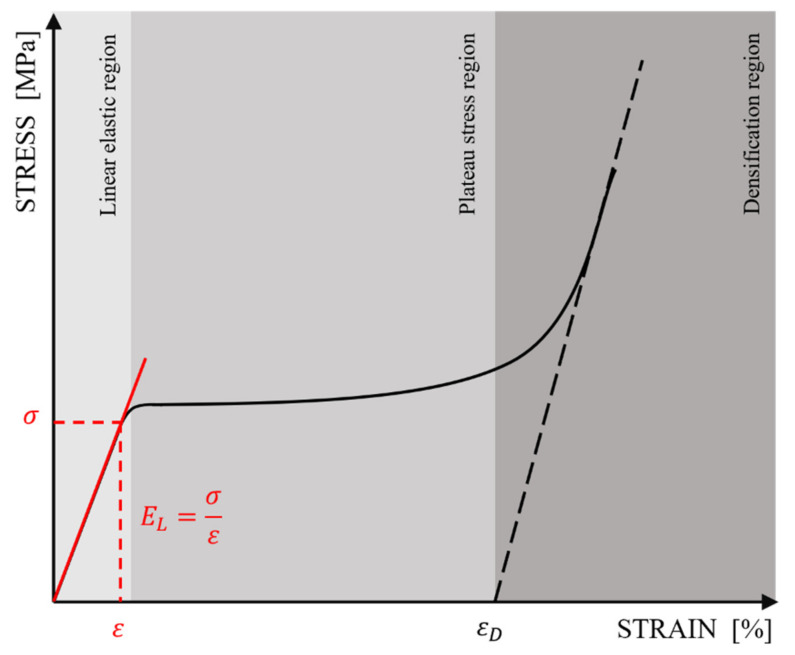
Theoretical compressive stress–strain curve for the cellular structure of plastic materials.

**Figure 2 materials-15-04352-f002:**
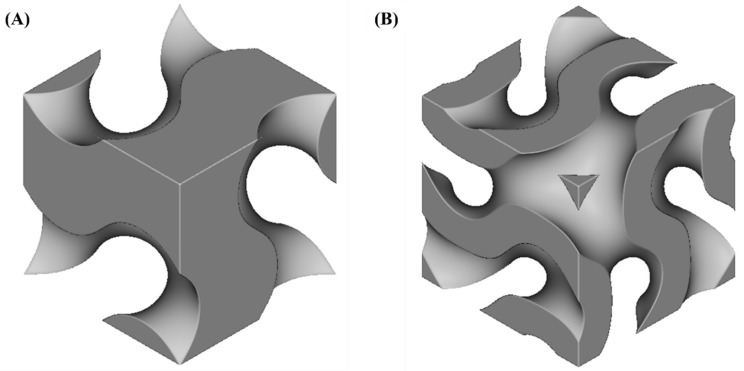
Gyroid unit cell: (**A**) network; (**B**) matrix (shell).

**Figure 3 materials-15-04352-f003:**
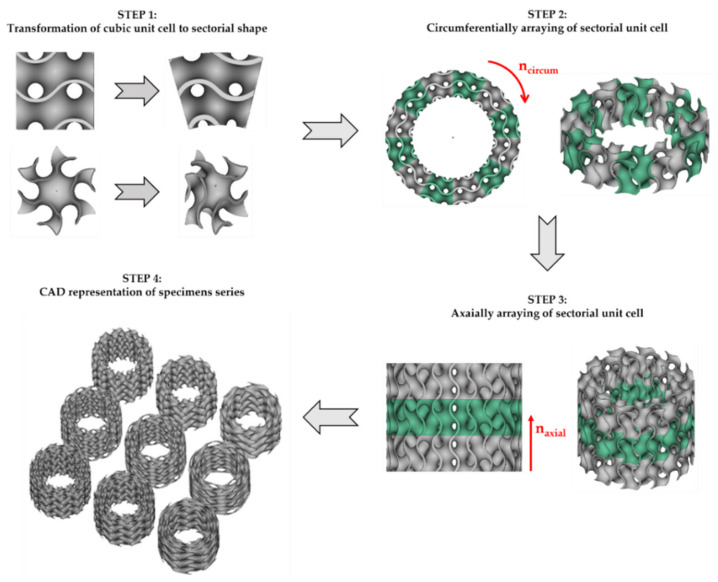
The design process of cylindrically mapped TPMS structures with shell gyroid unit cells.

**Figure 4 materials-15-04352-f004:**
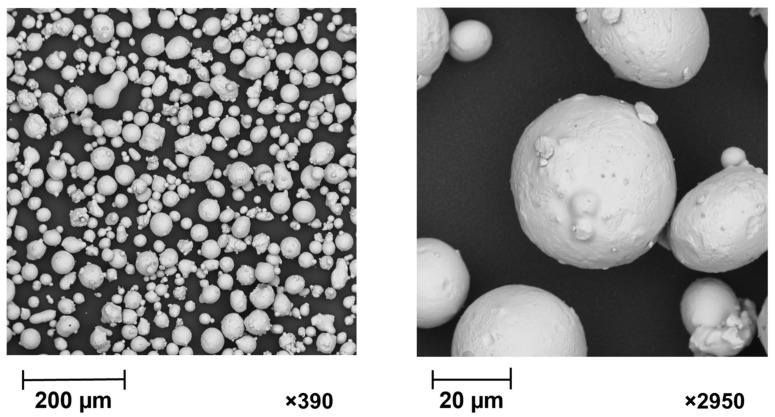
SEM image of 316L austenitic stainless steel powder MetcoAddTM 316L-A. The image shows the spherical morphology of the powder by magnitude, respectively, 390 and 2950, produced by the gas atomization process.

**Figure 5 materials-15-04352-f005:**
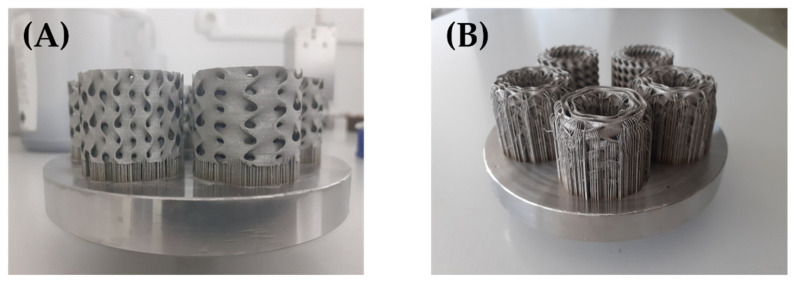
Comparison of support strategies: (**A**) for samples with n_radial_ = 1 and n_radial_ = 2; (**B**) for samples with n_radial_ = 1.5.

**Figure 6 materials-15-04352-f006:**
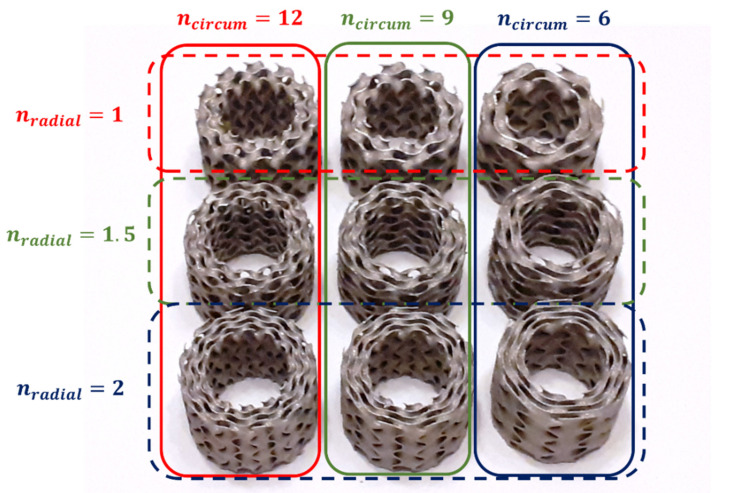
Cylindrical mapped TPMS structures with shell gyroid unit cells fabricated by selective laser melting (SLM) with 316L stainless steel used in the study.

**Figure 7 materials-15-04352-f007:**
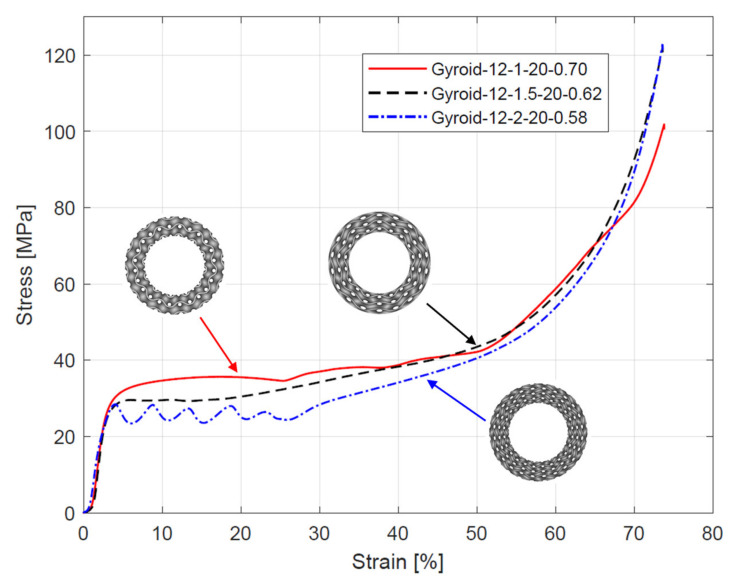
Stress–strain curves from compressive testing of cylindrically mapped gyroid structures with 12 unit cells in a circumferential direction.

**Figure 8 materials-15-04352-f008:**
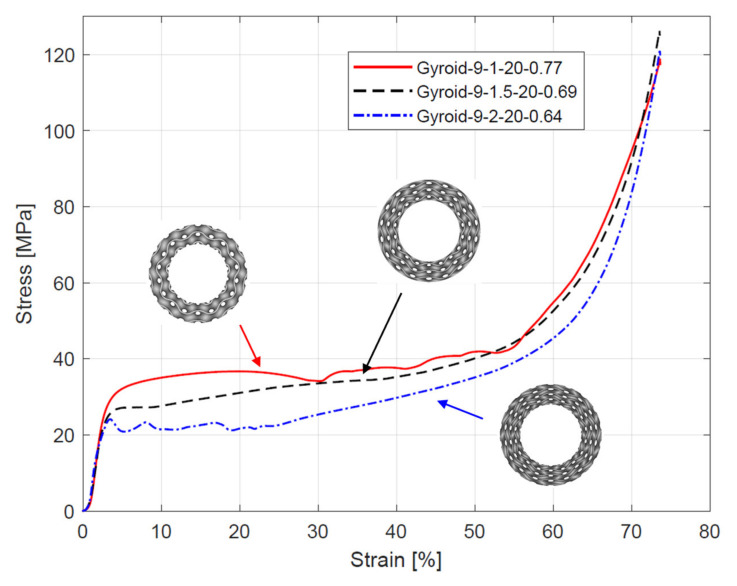
Stress–strain curves from compressive testing of cylindrically mapped gyroid structures with 9 unit cells in a circumferential direction.

**Figure 9 materials-15-04352-f009:**
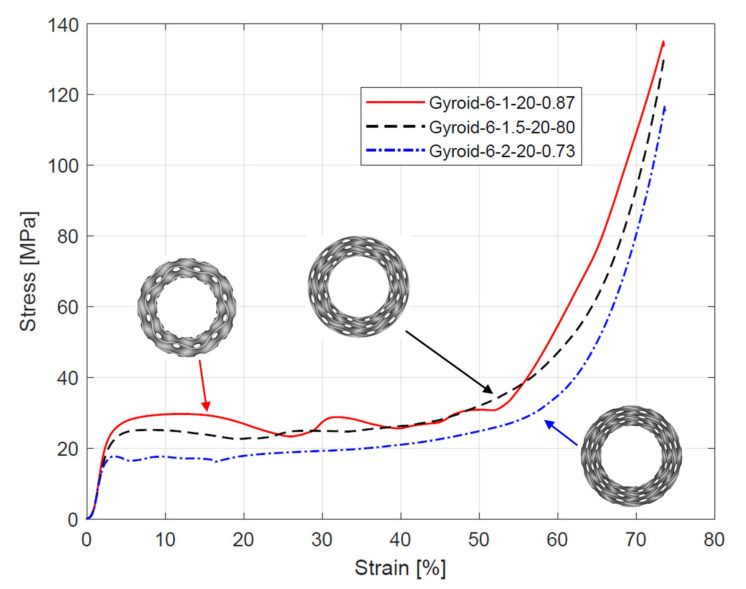
Stress–strain curves from compressive testing of cylindrically mapped gyroid structures with 6 unit cells in a circumferential direction.

**Figure 10 materials-15-04352-f010:**
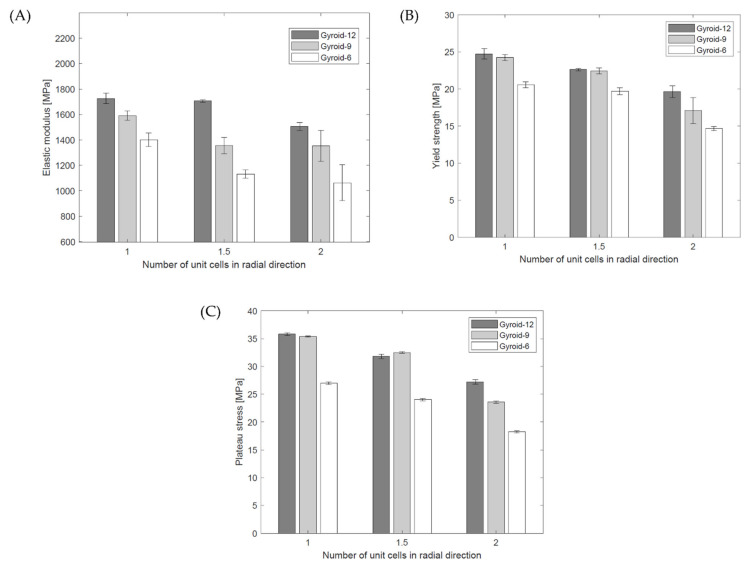
Mechanical properties of tested cylindrical mapped gyroid structures: (**A**) elastic modulus, (**B**) yield strength, (**C**) plateau stress.

**Figure 11 materials-15-04352-f011:**
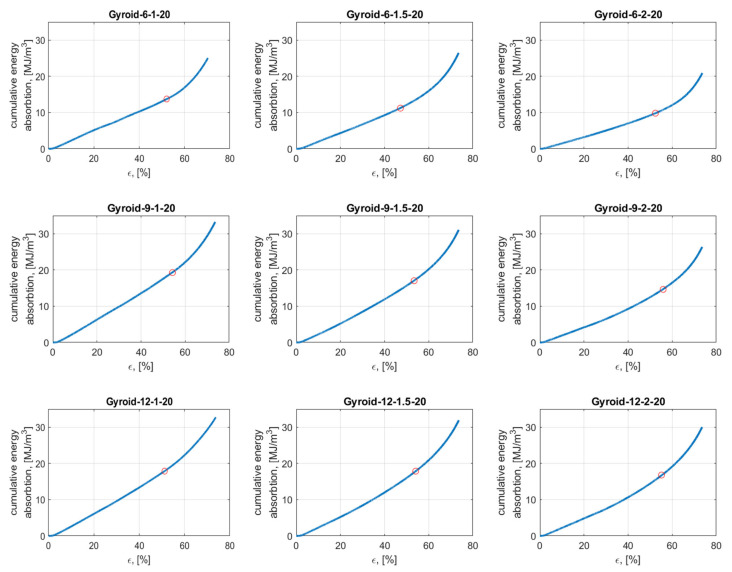
Energy absorption per unit volume of tested cylindrical mapped gyroid structures.

**Figure 12 materials-15-04352-f012:**
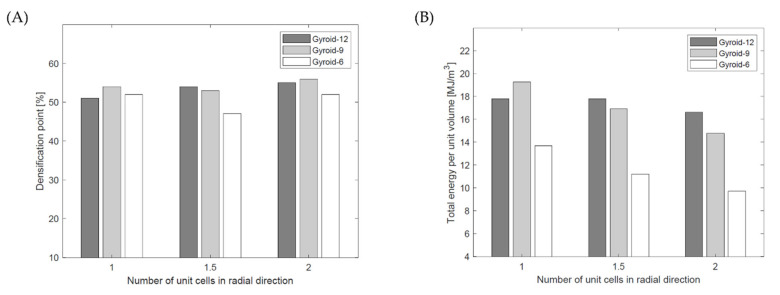
Densification point (**A**) and total energy absorbed per unit volume up to densification point (**B**) of tested cylindrical mapped gyroid structures.

**Figure 13 materials-15-04352-f013:**
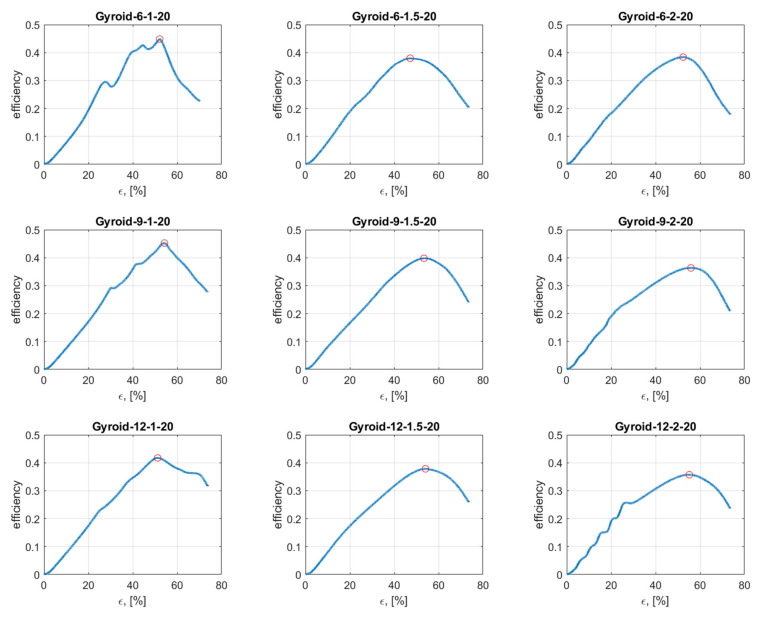
Efficiency–strain curve of tested cylindrical mapped gyroid structures.

**Figure 14 materials-15-04352-f014:**
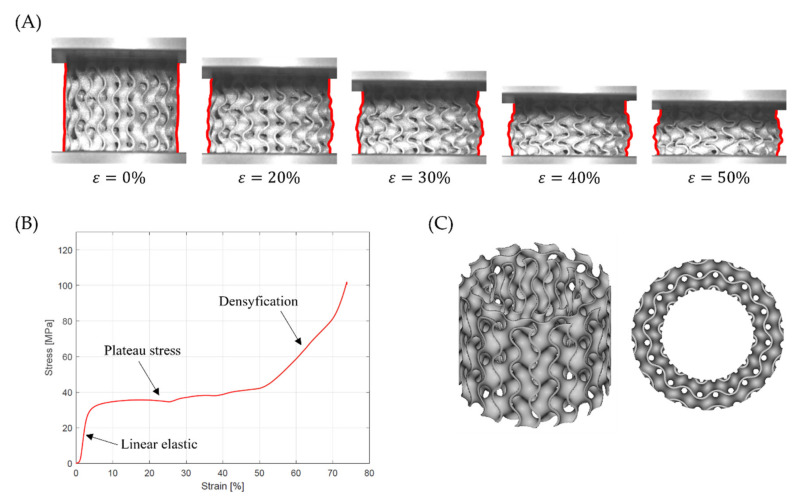
Global uniform deformation mode on the example of a Gyroid_12_1_20_0.70 samples: (**A**) deformation stages in the presence of 0%, 20%, 30%, 40%, 50% strain recorded by ARAMIS measurement system; (**B**) compressive stress–strain curve; (**C**) CAD project.

**Figure 15 materials-15-04352-f015:**
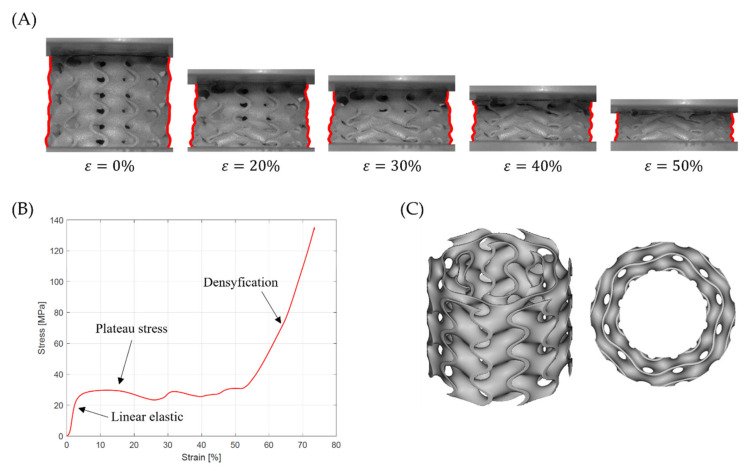
“Layer-by-layer” deformation mode with complete densification of collapsing layers on the example of Gyroid_6_1_20_0.87 samples: (**A**) deformation stages in the presence of 0%, 20%, 30%, 40%, 50% strain recorded by ARAMIS measurement system; (**B**) compressive stress–strain curve; (**C**) CAD project.

**Figure 16 materials-15-04352-f016:**
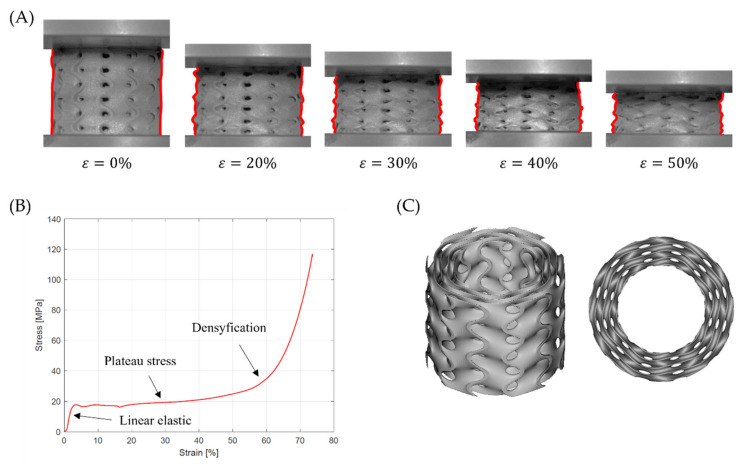
“Layer-by-layer” deformation mode with incomplete densification of collapsing layers on the example of Gyroid_6_2_20_0.73 samples: (**A**) deformation stages in the presence of 0%, 20%, 30%, 40%, 50% strain recorded by ARAMIS measurement system; (**B**) compressive stress–strain curve; (**C**) CAD project.

**Figure 17 materials-15-04352-f017:**
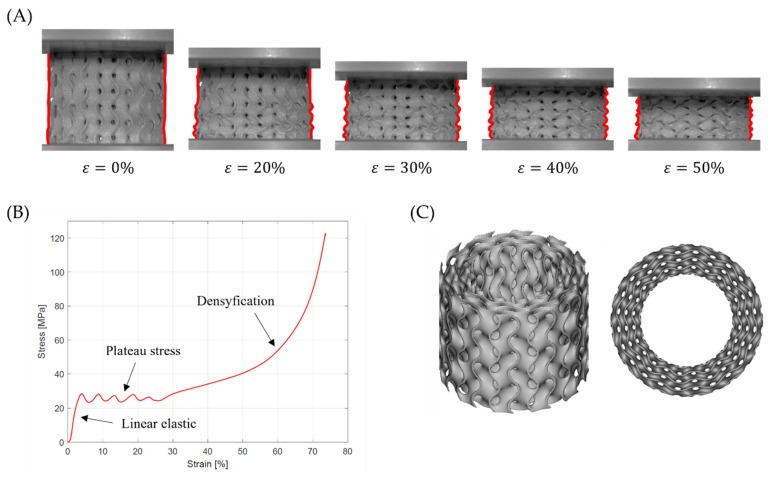
Special case of “layer-by-layer” deformation mode for Gyroid_12_2_20_0.58 samples: (**A**) deformation stages in the presence of 0%, 20%, 30%, 40%, 50% strain recorded by ARAMIS measurement system; (**B**) compressive stress–strain curve; (**C**) CAD project.

**Table 1 materials-15-04352-t001:** Chemical composition of MetcoAdd^TM^ 316L-A powder [42].

Element	Fe	Cr	Ni	Mo	C	Other
**Weight percent [%]**	Balance	18	12	2	<0.03	<1.0

**Table 2 materials-15-04352-t002:** Manufacturing parameters of the SLM process.

Laser Power	Laser Speed	Layer Thickness	Printing Environment
123 W	1000 mm/s	25 µm	Argon

**Table 3 materials-15-04352-t003:** Specifications of samples used in this study.

Symbol	n_circum_	n_radial_	n_axial_	T [mm]	m_r_ [g]	m_r_avrage_ [g]	Δm [%]
Gyroid_12_1_20_0.70					16.72	16.56	0.63
	12	1	3	0.70	16.52		
					16.44		
Gyroid_12_1.5_10_0.62					16.26	16.32	−0.83
	12	1.5	3	0.62	16.28		
					16.34		
Gyroid_12_2_20_0.58					16.33	16.41	−0.28
	12	2	3	0.58	16.46		
					16.45		
Gyroid_9_1_20_0.77					16.72	16.60	0.87
	9	1	3	0.77	16.73		
					16.86		
Gyroid_9_1.5_20_0.69					16.36	16.36	−0.63
	9	1.5	3	0.69	16.23		
					16.48		
Gyroid_9_2_20_0.64					16.32	16.50	0.22
	9	2	3	0.64	16.72		
					16.45		
Gyroid_6_1_20_0.87					16.69	16.51	0.28
	6	1	3	0.87	16.45		
					16.38		
Gyroid_6_1.5_20_0.80					16.24	16.32	−0.85
	6	1.5	3	0.80	16.24		
					16.28		
Gyroid_6_2_20_0.73					16.47	16.58	0.71
	6	2	3	0.73	16.58		
					16.68		

**Table 4 materials-15-04352-t004:** Elastic modulus, yield strength, and plateau stress of the tested cylindrical mapped gyroid structures.

Symbol	Elastic Modulus [MPa]	Yield Strength [MPa]	Plateau Stress [MPa]
Gyroid_12_1_20_0.70	1726.16 ± 42.35	24.72 ± 0.69	35.80 ± 0.25
Gyroid_12_1.5_20_0.62	1706.33 ± 8.69	22.61 ± 0.16	31.81 ± 0.38
Gyroid_12_2_20_0.58	1504.81 ± 32.47	19.63 ± 0.80	27.20 ± 0.39
Gyroid_9_1_20_0.77	1590.91 ± 37.11	24.22 ± 0.40	35.39 ± 0.12
Gyroid_9_1.5_20_0.69	1355.98 ± 64.50	22.43 ± 0.40	32.51 ± 0.16
Gyroid_9_2_20_0.64	1353.19 ± 121.480	17.09 ± 1.77	23.57 ± 0.18
Gyroid_6_1_20_0.87	1401.53 ± 53.57	20.57 ± 0.40	27.00 ± 0.23
Gyroid_6_1.5_20_0.80	1130.34 ± 33.24	19.68 ± 0.46	24.02 ± 0.19
Gyroid_6_2_20_0.73	1063.36 ± 141.04	14.67 ±0.27	18.24 ± 0.20

**Table 5 materials-15-04352-t005:** Densification point and total energy per unit volume absorbed up to densification point of the tested cylindrical mapped gyroid structures.

Symbol	Densification Point [%]	Total Energy Per Unit Volume [MJ/m^3^]
Gyroid_12_1_20_0.70	51	17.80
Gyroid_12_1.5_20_0.62	54	17.78
Gyroid_12_2_20_0.58	55	16.63
Gyroid_9_1_20_0.77	54	19.26
Gyroid_9_1.5_20_0.69	53	16.90
Gyroid_9_2_20_0.64	56	14.76
Gyroid_6_1_20_0.87	52	13.68
Gyroid_6_1.5_20_0.80	47	11.21
Gyroid_6_2_20_0.73	52	9.72

**Table 6 materials-15-04352-t006:** Classification of the tested samples to the observed deformation modes.

Gyroid_6_1_20_0.87	Gyroid_6_1.5_20_0.80	Gyroid_6_2_20_0.73
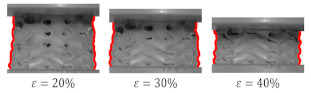	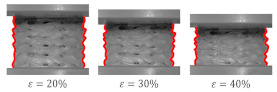	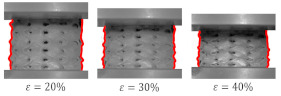
“layer-by-layer” deformation mode with complete densification of collapsing layers W_V_ = 13.68 [MJ/m^3^]; ε_D_ = 52% Appendix A	“layer-by-layer” deformation mode with incomplete densification of collapsing layers W_V_ = 11.21 [MJ/m^3^]; ε_D_ = 47%(min) Appendix A	“layer-by-layer” deformation mode with incomplete densification of collapsing layers W_V_ = 9.72 [MJ/m^3^] (min); ε_D_ = 52% Appendix A
**Gyroid_9_1_20_0.77**	**Gyroid_9_1.5_20_0.69**	**Gyroid_9_2_20_0.64**
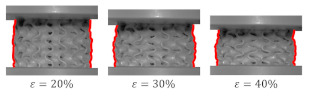	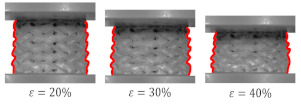	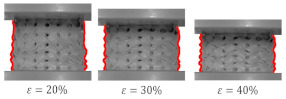
global uniform deformation modeW_V_ = 19.26 [MJ/m^3^] (max); ε_D_ = 54% Appendix A	“layer-by-layer” deformation mode with incomplete densification of collapsing layers W_V_ = 16.90 [MJ/m^3^]; ε_D_ = 53% Appendix A	“layer-by-layer” deformation mode with incomplete densification of collapsing layers W_V_ = 14.76 [MJ/m^3^]; ε_D_ = 56% Appendix A
**Gyroid_12_1_20_0.70**	**Gyroid_12_1.5_20_0.62**	**Gyroid_12_2_20_0.58**
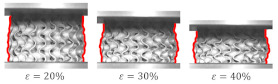	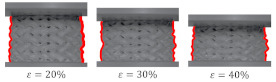	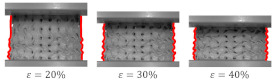
global uniform deformation mode W_V_ = 17.80 [MJ/m^3^]; ε_D_ = 51% Appendix A	“layer-by-layer” deformation mode with incomplete densification of collapsing layers W_V_ = 17.78 [MJ/m^3^]; ε_D_ = 54% Appendix A	special case of “layer-by-layer” deformation mode W_V_ = 16.63 [MJ/m^3^]; ε_D_ = 55% Appendix A

## Data Availability

Not applicable.

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
