# Peer review of "The Influence of the Structure Parameters on the Mechanical Properties of Cylindrically Mapped Gyroid TPMS Fabricated by Selective Laser Melting with 316L Stainless Steel Powder"

_materials, 2022, doi:10.3390/ma15124352_

Round 1

Reviewer 1 Report

The current manuscript investigates the effect of the structure parameters on the mechanical properties of cylindrically mapped gyroid TMPS from 316L stainless steel fabricated by SLM. The title is attractive and the presented results are interesting. However, some issues should be considered as follows:

  • The introduction is well written, however, there should be a detailed define of the problem statement  by the end of this section, and the direct applications of the recommended design shapes. 
  • There is no need for the introduction presented at the beginning of each section, such as section 2 and 3.
  • On Figure 4, the scale of images should be clear and adjusted. 
  • On Figures 8, 9, 10, the line styles should be varied to be compatible for easy discrimination on the black and white copies.
  • Standard deviation should be added to the data presented on Figure 11.
  • Figures 12, 14 are recommended to be divided into more figures to be clearly accessed and presented. 
  • To get the impact of the current work, a summary of the results value should be added to a separate table and compared to the results obtained from the literature studies. 
  • The conclusion section should focus on the main results, novelty, and significant contributions of the current work. A pullet points style is recommended. 

Author Response

Dear Reviewer,

Thank you very much for your comments and suggestion. In the attachment you will find a file with detailed responses.

Best regards, 

Authors

Reviewer 2 Report

Considering that this work is a review I think it would be very good if it is reorganized.  Maybe it would be advisable to introduce a chart with statistics about the fild. I don't think there are enough bibliographic notes. The goal of this research is not obvious. What they authors are trying to do and how is different from previous research is not mentioned anywhere. As far as I notice this paper would rather be an article than a review and consequently for each section should be treated as such as a result I do not agree with its publication.

Author Response

(The authors gave the same response as above.)

Reviewer 3 Report

The research presents the mechanical responses of porous structures with various structure parameters. Overall the design of structures, mechanical testing and data collection are OK, but the work is more like an experimental report which simply presents the obtained results and lacks adequate discussion to make the work useful to other researchers. Therefore, I think more discussion of the results should be added.

1. The authors found three deformation modes of the structures according to the stress-strain curves, but did not compare between them to show the advantages and disadvantages of each mode. Which deformation mode is preferred and which should be avoided in industry? Does the preference of deformation mode vary with application scenarios? These questions should be discussed.

2. According to the work, the deformation mode varies with structure parameters. But what is the root cause of such deformation mode transition? In materials science, it is very important to reveal the mechanisms of observed phenomena from a structure point of view. The authors should try to study the underlying structural properties that govern deformation mode.

3. Although the authors claim that the work aims for more efficient structure design, I cannot see how other researchers can benefit from this work. Based on the experimental data, the authors should try to reach some more generalized conclusions that can be extended, such as quantitative models or at least some qualitative descriptions or suggestions in terms of structural design.

There are also some other wording problems that should be carefully checked. E.g., the authors seem to mix ‘deformation mode’ and ‘deformation model’, which should be unified.

Author Response

(The authors gave the same response as above.)

Round 2

Reviewer 1 Report

The revised manuscript is significantly improved. There is a minor issue to be consider; the conclusion section should be focused, there is no need to refer to any references or figures and please avoid data redundancy.

Author Response

Dear Reviewer,

Thank you very much for your comments and suggestions.

Best regards,

Authors

Reviewer 2 Report

I agree with the changes made. The article can be published.

Author Response

(The authors gave the same response as above.)

Reviewer 3 Report

The manuscript has been improved after the amendments. I have no further comments.

Author Response

(The authors gave the same response as above.)
